# Establishment of a New Real-Time Molecular Assay for the Detection of Babanki Virus in Africa

**DOI:** 10.3390/v16121841

**Published:** 2024-11-27

**Authors:** Martin Faye, Mathilde Ban, Fatou Kiné Top, El Hadji Ndiaye, Fatou Diène Thiaw, Gamou Fall, Moussa Moise Diagne, Amadou Alpha Sall, Mawlouth Diallo, Valérie Choumet, Ousmane Faye

**Affiliations:** 1Virology Department, Institut Pasteur de Dakar, 36 Avenue Pasteur, Dakar 220, Senegal; fatoukine.top@pasteur.sn (F.K.T.); fatoulayethiaw25@gmail.com (F.D.T.); gamou.fall@pasteur.sn (G.F.); moussamoise.diagne@pasteur.sn (M.M.D.); amadou.sall@pasteur.sn (A.A.S.); ousmane.faye@pasteur.sn (O.F.); 2Environment and Infectious Risk Unit, Institut Pasteur, Université Paris Cité, 75724 Paris, France; mathilde.ban@pasteur.fr (M.B.); valerie.choumet@pasteur.fr (V.C.); 3Zoology Department, Institut Pasteur de Dakar, 36 Avenue Pasteur, Dakar 220, Senegal; elhadji.ndiaye@pasteur.sn (E.H.N.); mawlouth.diallo@pasteur.sn (M.D.)

**Keywords:** Babanki virus, molecular diagnostic, *q*RT-PCR, Africa

## Abstract

Babanki virus is a subtype of the Sindbis virus, a widespread arthropod-borne alphavirus circulating in Eurasia, Africa, and Oceania. Characterized by rashes and arthritis, clinical infections due to Sindbis were mainly reported in Africa, Australia, Asia, and Europe. However, its sub-type, Babanki virus, was reported in Northern Europe and Africa, where its epidemiology potential remains poorly understood. The diagnosis of alphaviruses is mainly based on serological testing and conventional PCR methods, which have considerable limits. In this study, we developed a real-time *q*RT-PCR assay for the detection of Babanki virus. The analytical sensitivity and specificity of the newly established assay were evaluated using in vitro standard RNA and related viruses relevant to the African context, respectively. In addition, its diagnostic sensitivity was assessed using a subset of Babanki virus-positive and -negative mosquito pools collected from the field. The new real-time *q*RT-PCR assay exhibited a 100% specificity, a 95% detection limit of 1 RNA molecule/reaction, and a diagnostic sensitivity of up to 120 pfu/reaction. This newly established assay could be useful not only for the detection of Babanki virus during epidemics but also in future experimental and surveillance studies focusing on their epidemiology and pathogenicity.

## 1. Introduction

Arthropod-borne pathogens have contributed significantly to the emergence and re-emergence of infectious diseases in recent decades [1]. Alphaviruses constitute an important group (>30 species of linear, single-stranded, positive-polarity RNA viruses) among these pathogens capable of infecting humans and animals [2]. They can be divided into two groups: Old World Alphaviruses, whose prototype is Sindbis virus (SINV), responsible for arthritis and skin rashes; and New World Alphaviruses responsible for encephalitis and divided into two complexes, Eastern Equine Encephalitis (EEEV) and Venezuelan Equine Encephalitis (VEEV) [3]. Belonging to the Togaviridae family, some African alphaviruses, such as the Sindbis virus (SINV), are transmitted by mosquito vectors and cause disease in humans [4].

SINV is an arthritogenic alphavirus, known to spontaneously cause acute febrile illness (AFI) in Africa, Australia, Asia, and Europe. Epidemics are associated with heavy rainfall and temperature changes that favor mosquito breeding. SINV was identified as a cause of human disease in South Africa in 1963, and subsequent studies confirmed that the virus was present in mosquito populations in the Central Plateau region, which includes the Free State province [4]. It was first isolated in 1952 from *Culex pipiens* and *Culex univittatus* collected in Sindbis, a village then north of Cairo, Egypt. Its main reservoir hosts are birds, and its vectors are mosquitoes of the *Culex* (*Culex torrentium*, *Culex univittatus*, *Culex pipiens*, *Culex quinquefasciatus*), *Anopheles* (*Anopheles funestus*, *Anopheles gambia*), *Aedes* (*Aedes mcintoshi*, *Aedes communis*, *Aedes punctor*, *Aedes diantaeus*, *Aedes albopictus*, *Aedes aegypti*, *Aedes sticticus*, *Aedes koreicus*, *Aedes geniculatus*) and *Culiseta* (*Culiseta morsitans*) genera. Although six geographically distinct genotypes of SINV have been previously reported, only the SINV-I genotype has been associated with infections in humans [5]. Depending on the location, the disease is sometimes referred to as Babanki, Ockelbo, Karelian fever, Kyzylagach, or Whataroa [6].

While SINV has confirmed its potential to cause widespread epidemics in humans, the epidemic potential of its subtype Babanki virus (BBKV) remains uncertain [7]. BBKV is 98% similar to European SINV strains based on nucleotide sequences [8]. BBKV could be transmitted by *Culex quinquefasciatus*, *Culex neavei*, and *Lutzia tigripes* [5]. It was first isolated in 1969 in the Babanki forest in Cameroon from the species *Mansonia africana* and has since been isolated from numerous other species of the genera *Aedes*, *Culex*, *Eretmapodites*, and *Lutzia* [9], in northern Europe and Africa [10]. BBKV is widespread in the Afrotropical region, including Madagascar, and has been detected in rodents and bats, but it is above all a virus that affects birds and can infect humans. It then causes fever, arthralgia, joint pain, and rash in humans. In Senegal, BBKV has been previously isolated from two mosquito pools collected through a surveillance program for arboviruses [9]. The laboratory diagnosis of human alphavirus infections has changed greatly over the last few years. In the past, the identification of alphavirus relied on four tests involving the detection of antibodies: hemagglutination inhibition, complement fixation, plaque reduction neutralization test, and the indirect fluorescent antibody (IFA) test. Positive identification using these immunoglobulin M (IgM)- and IgG-based assays required a fourfold increase in titer between acute and convalescent serum samples [11,12]. However, these methods are relatively time-consuming and far less sensitive than new molecular biology methods such as PCR. Due to the high frequency of travel to endemic regions, the wide variety of reservoirs, and the potential for vector-borne spread, the development of a sensitive and specific pan-alphavirus assay is challenging. Nevertheless, PCR-based assays have been previously described for the detection of major alphaviruses such as Chikungunya virus (CHIKV) and Eastern [13], Western [14], and Venezuelan [15] equine encephalitis viruses (EEE, WEE, and VEE, respectively). However, the specificity of these assays towards the closely related heterologous alphaviruses was low [16]. Thus, the establishment of specific new molecular tools targeting emerging alphaviruses such as BBKV is necessary for their rapid identification through surveillance programs.

Herein, we report on the development and validation of a specific and sensitive *q*RT-PCR assay for the detection of BBKV. Clinical performances of the newly established assay were evaluated using spiked human serum, cell-culture medium, and mosquito pools directly collected from the field.

## 2. Materials and Methods

### 2.1. Ethical Statement

The mosquito pools used in this study were collected as part of a national integrated surveillance program for arboviruses in Senegal and were obtained from the collection of the WHO Collaborating Center for Arboviruses and Haemorrhagic Fevers (CRORA) at the Institut Pasteur de Dakar (IPD), accredited for routine diagnosis and identification of pathogens under the IACUC guidelines.

### 2.2. Sample Collection

A stock of BBKV was obtained from the WHO collection at the Collaborating Center for Arboviruses and Viral Hemorrhagic Fevers (CRORA), Institut Pasteur de Dakar (IPD), Senegal.

Virus stocks of other arboviruses relevant in the African context (chikungunya virus, west Nile virus, dengue 1 and 2 virus, rift valley fever virus, and yellow fever virus) were used to assess the diagnostic specificity of the newly established assay. In addition, a total of 18 mosquito pools collected from the field in Senegal, between 2016 and 2020, were used to assess diagnostic sensitivity. BBKV-positive human serum samples collected in Senegal were also tested for analytical sensitivity.

### 2.3. Virus Stock Preparation

A viral stock of BBKV was prepared by inoculating Culex cell lines with infected chicken blood. Then, C6/36 mosquito cells were cultured in growth medium in 25 cm^2^ (T25) flasks and incubated at 28 °C in an oven for 24 h prior to infection, so as to have a cell mat confluent to around 80%. After pouring in the growth medium the previous day, 100 µL of viral stock was added to one of the pre-incubated T25 dishes. The cells were then incubated at 28 °C for 1 h in an oven at 28 °C to allow direct virus–cell contact. After this incubation time, 5 milliliters of survival medium were added to the dish and incubated again in the 28 °C oven until a cytopathic effect (CPE) was observed. The supernatant was harvested and stored at −80 °C, then tested by RT-PCR and plaque assay. Cells were analyzed for infectivity by indirect immunofluorescence.

### 2.4. Indirect Immunofluorescence (IFI)

Indirect immunofluorescence is a serological technique based on the detection of viral proteins present on the surface of infected cells using specific antibodies coupled to fluorescein. Once harvested, the cells were recovered by centrifugation of the supernatant and resuspended in 1× PBS (Phosphate Buffered Saline). The cells were then placed in dedicated wells on an immunofluorescence slide and put in the oven to dry. After drying, the cells were placed in cold acetone at −20 °C, then dried again. Specific primary antibody (ascites immune BBKV) diluted 20-fold was then placed in contact with the cells and incubated at 37 °C to dry for 30 min. This was followed by 3 washes with PBS 1× for 5 min each. After drying at room temperature, the cells were incubated at 37 °C in the presence of 80-fold diluted secondary antibody, to which Evans blue diluted 1/100 in PBS was added. After 30 min incubation for drying and 3 successive washes of 5 min each, a drop of glycerol (immersion liquid) was added to the slide and covered with a coverslip for reading the results under a luminescent microscope.

### 2.5. Plaque Assay

BBKV stock was titrated by the range method using Vero cells (monkey kidney cells). For this, cultured and pre-incubated Vero cells were infected in 24-well plates with dilutions of 10^−^^1^ to 10^−^^11^ of the sample to be titrated. In each well, 200 µL of L-15 supplemented with 5% fetal bovine serum (FBS) and 200 µL of suspension were added. The cells were then infected with 200 µL dilutions of the viral stock. After 4 h of incubation at 37 °C, 400 µL of overlay medium (50% carboxymethyl cellulose (CMC) and 50% L-15 with 5% FBS) was added to each well, and the plate was incubated again at 37 °C. After 3 days, the medium was removed, and the cells were washed with 1× PBS and stained with amido black for 30 min. After 30 min, the plates were rinsed with water and dried. The viral titer, expressed as plate-forming unit (pfu)/mL, was calculated using the last dilution of plate-forming virus (respecting the dilution factor) according to the Reed–Muench method:Titer (in pfu/mL) = Average number of lysis patches × Dilution factor × 5

### 2.6. Primers Design

Multiple alignments of BBKV nucleotide sequences available online (www.ncbi.nlm.nih.gov/genbank/, accessed on 25 May 2024) were carried out by using Muscle algorithm [17] within Unipro UGENE software version 39 [18]. The nsP3 protein’s alphavirus unique domain (AUD) is highly conserved among members of the alphavirus genus and has been found suitable for the design of specific primers to all alphaviruses [19]. Both primers and inverse-sense TaqMan probes were designed on the nsP3 protein using Primer3web^®^ software (version 4.0.0, Whitehead Institute for Biomedical Research, Cambridge, MA, USA) and submitted to a BLAST analysis on NCBI to avoid non-specific cross-reactions. Primers were synthetized by TIBMol-Biol (Berlin, Germany) and details are summarized in Table 1.

### 2.7. RNA Extraction

Viral RNA extraction from 140 µL of virus stocks or 10-fold serial dilutions of L-15 medium samples and human serum was performed with the QIAamp viral RNA mini kit (Qiagen, Heiden, Germany) according to the manufacturer’s instructions. Viral RNA was eluted in a final volume of 60 μL and frozen at −80 °C prior to downstream applications. As a Sindbis virus, Babanki virus is a Biosafety level 2 pathogen and molecular testings were performed in a biosafety level 2 laboratory at the Institut Pasteur de Dakar (IPD).

### 2.8. Real-Time qRT-PCR Conditions

Real-time *q*RT-PCR was performed using the *q*Script One-step *q*RT-PCR kit (Quanta Biosciences, Gaithersburg, MD, USA) in a final volume of 20 μL following the established protocol, and the reaction was carried out on a 7500 Fast Real-Time system cycler (Applied Biosystems, Foster City, CA, USA).

### 2.9. Specificity Testing

To assess the analytical specificity of the new BBKV RT-*q*PCR assays, RNA samples from 2 BBKV-positive isolates and 7 isolates of other arboviruses were tested. Positive and negative controls containing positive RNA and nuclease-free water, respectively, were included in each assay. In addition, a total of 12 flavivirus-positive mosquito pools (*Barkedji virus* and *Usutu virus*) were also analyzed using the *q*RT-PCR assay to determine diagnostic specificity. All samples were tested in duplicate.

### 2.10. Sensitivity Testing

#### 2.10.1. Analytical Sensitivity

The detection limit of the *q*RT-PCR assay for newly established BBKV was assessed using data from the 5 replicates over a range of in vitro RNA standard dilutions (10^6^ to 1 RNA molecule/reaction). Linear regression analysis and probit regression analysis with 95% probability were performed using data from 5 assays. Graphs were plotted using PRISM v10.2.2 (GraphPad Software Inc., San Diego, CA, USA).

#### 2.10.2. Diagnostic Sensitivity

RNAs extracted from the 5 pools of SINV-positive mosquitoes and 13 pools of BBKV and SINV-negative mosquitoes by in vitro isolation [20] were tested by *q*RT-PCR to confirm the reliability of the newly established tests.

#### 2.10.3. Sensitivity in Human Serum and Leibovitz 15 (L-15) Medium

Serial 10-fold dilutions of BBKV virus stock with a known titer were prepared in Leibovitz 15 (L-15) and human serum to assess the diagnostic sensitivity of the newly developed *q*RT-PCR assay. All dilutions were tested in triplicate. A linear regression curve was obtained from 10 *q*RT-PCR data sets of 10-fold dilutions of the molecular standard RNA. The generated equation was used to calculate the number of RNA molecules from the threshold cycle (Ct) values.

#### 2.10.4. Repeatability and Reproducibility

RNA extracted from BBKV stock with one titer was analyzed 8 times within the same assay and in 8 different assays to determine intra-assay and inter-assay coefficients of variation (CV).

#### 2.10.5. Experimental Validation Using Vector Competency

To assess its sensitivity for the detection of Babanki virus in various organs of mosquitoes and its usefulness for experimental studies, the newly established assay was validated using vector competency of Senegalese *Aedes aegypti*, *Culex neavei*, and *Culex quinquefasciatus* for BBKV as previously described [21]. BBKV infection rates were measured by the number of infected mosquito bodies for each species. Overall, we found a significant effect on post-infection days and infection rates when analyzing the 3 species. BBKV dissemination rates for the *Culex quinquefasciatus* population were measured by the number of mosquitoes with positive legs per 100 mosquitoes infected. Experimental study was conducted in a biosafety level 3 insectarium facility at the IPD.

### 2.11. Statistical and Data Analysis

To assess the sensitivity and specificity of the new *q*RT-PCR assay for BBKV, various statistical methods were applied. Linear regression was used to assess the relationship between threshold cycles (Ct) and the number of RNA molecules detected per reaction. This approach confirmed the assay’s ability to efficiently detect low RNA concentrations. A probit regression analysis was applied to determine the detection limit at 95% probability, which is essential for assessing the test’s robustness and reliability under diagnostic conditions.

For performance assays, analytical sensitivity was calculated from multiple replicates using serial dilutions of RNA standards; diagnostic sensitivity was obtained by comparing results obtained with other methods such as conventional RT-PCR, in vitro isolation, and intra- and inter-assay variability, and by calculating coefficients of variation (CVs) to ensure reproducibility of results.

The following software packages were used for data analysis and visualization: PRISM v10.2.2 (GraphPad) was used to plot regression curves and statistical analyses, providing clear and accurate graphs; Primer3web version 4.1.0 and Unipro UGENE version 39 were used to design primers and probes, ensuring their specificity through BLAST analysis.

To assess vector competence, infection rates of mosquito populations were evaluated using distribution models. These models made it possible to compare infection and dissemination rates between different mosquito species, with implications for epidemiology and epidemic surveillance.

## 3. Results

### 3.1. Specificity

No amplification was observed for any no-BBKV strain, while the two BBKV-positive mosquito pools were detected using the qRT-PCR assay, resulting in a diagnostic specificity of 100% for the newly established BBKV qRT-PCR assay (Table 2).

### 3.2. Data from the Analytical Sensitivity Testing

The analytical sensitivity of the new *q*RT-PCR assay for newly established BBKV was determined with Ct data values from six sets of one-tenth dilutions of the in vitro RNA standard ranging from 10^6^ to 1 molecule/reaction. The BBKV *q*RT-PCR assay detected the in vitro RNA standard with a concentration from 10^6^ to 1 molecule/reaction in six assays (Figure 1A). With these data sets, a probit regression analysis was performed and revealed a detection limit with a 95% probability of 1 RNA molecule/reaction for the newly established *q*RT-PCR assay. The assay cut-off for the Ct value is 39.8 (Figure 1B). Intra-series CVs of 0.27 ± 0.058 and inter-series CVs of 0.05 ± 0.075 were found for the newly established *q*RT-PCR assay, indicating that these tests are reproducible.

### 3.3. Data from the Diagnostic Sensitivity Testing

All mosquito samples collected were analyzed by *q*RT-PCR in comparison to data from the in vitro isolation and the conventional pan-alphavirus RT-PCR testing. The newly established BBKV *q*RT-PCR detected 100% of RT-PCR-positive samples (7/7) and 100% of isolation positive samples (5/5). In addition, it detected 69.2% (9/13) of isolation negative samples and 53.4% (7/13) of RT-PCR negative samples for BBKV (Table 3).

### 3.4. Data from the Sensitivity Testing in Human Serum and Leibovitz 15 (L-15) Medium

One-tenth dilutions of a BBKV stock prepared in human serum and 10% L-15 medium were analyzed in triplicate using the BBKV-specific *q*RT-PCR assay. The new *q*RT-PCR assay gave a sensitivity of up to 120 pfu/reaction in both human serum and 10% L-15 medium (Figure 2A), corresponding to less than 10 RNA molecules/reaction calculated from Ct values and the equation obtained from the linear regression analysis of 10 *q*RT-PCR data sets of 10-fold dilutions of molecular standard RNA (Figure 2B).

### 3.5. Vector Competency of Mosquitoes for BBKV

The analysis of infection rates by species and post-infection days showed that *Aedes aegypti* exhibited 100% of infection at 7 and 15 days post-infection (ranging from 10^6^ to 10^4^ RNA molecules/reaction). However, *Culex neavei* and *Culex quiquefasciatus* showed infection ranging from 75% to 62.5% and from 37.5% to 62.5% (10^5^ to 10^2^ RNA molecules/reaction) after 7 and 15 days post-infection, respectively (Figure 3A,B). The dissemination rates in *Culex quinquefasciatus* ranged from 25% to 50% (10^4^ to 10^2^ RNA molecules/reaction) and from 40% to 50% (10^3^ to 10 RNA molecules/reaction) at 7 and 15 days post-infection, respectively (Figure 3A,B). The newly developed assay was then able to detect viral loads ranging from 10^6^ to 10 RNA molecules/reaction in different parts of the mosquito’s body.

## 4. Discussion

The epidemiology of alphaviruses remains poorly understood due to the sporadic nature of epidemics caused by these viruses. Nevertheless, alphaviruses have a wide geographical distribution in all continents, associated with their potential for dissemination through a number of mosquito species [22]. In addition, the changing patterns of infections due to alphaviruses point to the crucial need for improvement of diagnostic methods and strengthening preparedness and disease surveillance worldwide. Molecular methods have been widely used in epidemiological surveys. However, due to the large genetic diversity in the alphavirus genus, only a conventional nested pan-alphaviruses RT-PCR assay has been previously established [23]. Therefore, it is important to establish rapid and reliable tools for the surveillance of alphaviruses of public health concern in Africa where resources and access to diagnostics are limited. Herein, we developed a real-time *q*RT-PCR assay for the rapid detection of BBKV.

The newly established *q*RT-PCR assay detected all the SINV and BBKV-positive strains tested in our study, without any cross-reactivity with other related arboviruses relevant in the African context. Due to the high sequence similarity in the primer binding region of the nsP1 protein, the assay could be useful for the wide detection of SINV and BBKV strains, as previous genomic data showed that BBKV is a SINV subtype [24]. The high sensitivity of our *q*RT-PCR test for the detection of Babanki alphavirus has been demonstrated on the basis of its ability to detect as few as 10 RNA molecules/reaction. Similar analytical sensitivity has been previously reported for an assay targeting Venezuelan equine encephalitis virus [25]. In addition, a one-step TaqMan^®^ real-time RT-PCR assay detected all available Finnish SINV strains from cell culture supernatants with an analytical sensitivity of nine copies/reaction [26]. The advantages of using the real-time RT-PCR test (SYBR-Green) were also demonstrated for Chikungunya virus (CHIKV) detection, thanks to its sensitivity, speed (less than 3 h to complete the analysis) and accuracy. In addition, the test is quantitative, can be easily standardized with a very low detection limit of 4.12 × 100 RNA molecules/reaction, and presents a lower risk of contamination [27]. Our newly established assay exhibited also a high repeatability, as both inter-assay and intra-assay variability were lower than 1%. It could be, then, applied in surveillance studies as a differential testing method and a second line testing after a conventional pan-alphaviruses RT-PCR, for any suspicion of SINV and BBKV.

In addition, this new *q*RT-PCR assay exhibited a diagnostic sensitivity of up to 120 pfu/reaction in both human serum and 10% L-15 medium, corresponding to fewer than 10 RNA molecules/reaction. Previous data showing that acute SINV infection was generally associated with a low viral load < 103 RNA molecules/reaction in serum samples [26] imply that RT-PCR may not be suitable for the clinical diagnosis of SINV. However, our newly established assay is highly sensitive and could be useful not only in central laboratories receiving clinical specimens but also in field conditions for the testing of collected mosquito pools if coupled with portable PCR equipment. In addition, it could be useful for first-line testing and the screening of infections due to SINV and BBKV, during epidemics and in areas where virus circulation has been reported.

Vector competency data revealed high infection rates for BBKV in Senegalese populations of *Aedes aegypti*, *Culex neavei*, and *Culex quinquefasciatus* as well as a high dissemination rate in *Culex quinquefasciatus*. These rates were higher than those revealed by previous data from populations of *Culex quinquefasciatus* infected with CHIKV, which showed infection and dissemination rates of 25–31% and 13–40% at days 7–14 and day 21 post-infection, respectively [28]. In addition, our data showed also that Senegalese mosquito populations of the assessed *Culex* and *Aedes* genera could disseminate BBKV under laboratory conditions, as previously described for SINV [29]. These vectors could, therefore, contribute to sustained epidemics of Babanki virus in endemic areas. However, further experimental studies are needed to assess the ability of *Aedes aegypti*, *Culex neavei*, and *Culex quinquefasciatus* to transmit the virus. In addition, our new assay was highly sensitive for the detection of Babanki virus in various organs of mosquitoes and could be applied not only in future experimental studies but also in entomological surveillance programs focusing on Babanki virus.

BBKV has been associated with human febrile illness accompanied by rash and arthritis and the virus has been isolated from humans in Cameroon, Madagascar, and the Central African Republic. Although there has not been a large outbreak of BBKV in Africa yet, isolates of SINV and closely related BBKV have been recovered from *Culex* species from Naivasha, Kisumu, and Budalangi, neighboring sites to Lakes Naivasha and Victoria—Kenya (East Africa) [30]. These viruses were generally associated with migratory birds nesting around the lakes during their breeding season, which could be a risk factor for long-distance virus spread to new geographical regions, as previously observed for West Nile and avian influenza viruses, as well as for SINV in Australia [31]. In addition, previous data on the antigenic relatedness of alphaviruses have also suggested that their progenitor spread over long distances via birds’ migration [32] and phylogenetic studies have previously indicated a low genetic diversity between SINV strains isolated from South Africa and Northern Europe [33,34]. However, further integrated surveillance studies are needed to assess the role of local and migratory birds in the enzootic maintenance cycle of SINV and to better understand factors contributing to the epidemiological pattern of its circulation in Africa.

## Figures and Tables

**Figure 1 viruses-16-01841-f001:**
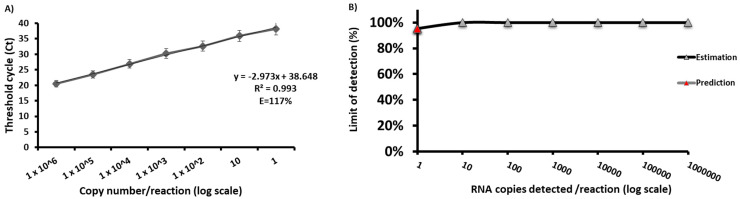
Analytical sensitivity of new *q*RT-PCR assay for BBKV. Linear regression analysis was performed by plotting *q*RT-PCR threshold cycle (Ct) values (**A**) against the number of RNA molecules per reaction detected in five replicates (5/5). Points represent mean values and error bars represent standard deviation. The *q*RT-PCR assay produced positive signals with dilutions from 10^6^ to 1 RNA molecule/reaction in five out of five analyses. Probit regression analysis was performed using data from the five *q*RT-PCR assays (**B**). Graphs were plotted using PRISM v10.2.2 (GraphPad Software Inc., San Diego, CA, USA), and the limit of detection with 95% probability is represented by the red triangle and is 1 RNA molecule/reaction.

**Figure 2 viruses-16-01841-f002:**
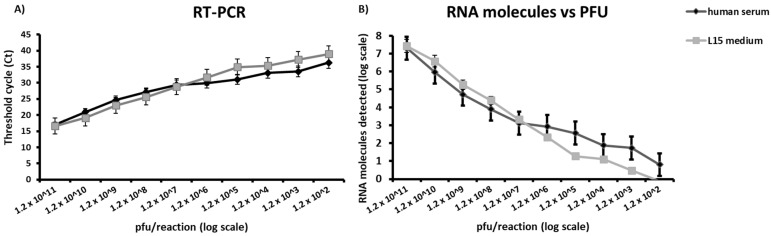
Analytical sensitivity of the newly established *q*RT-PCR assay using 10-fold serial dilutions of BBKV in human serum (black curve) and 10% L-15 medium (gray curve). Dilutions were tested in triplicate (**A**). The *q*RT-PCR assay detected up to 120 pfu/reaction, corresponding to 120 RNA molecules/reaction (**B**) using the equation obtained in Figure 1A.

**Figure 3 viruses-16-01841-f003:**
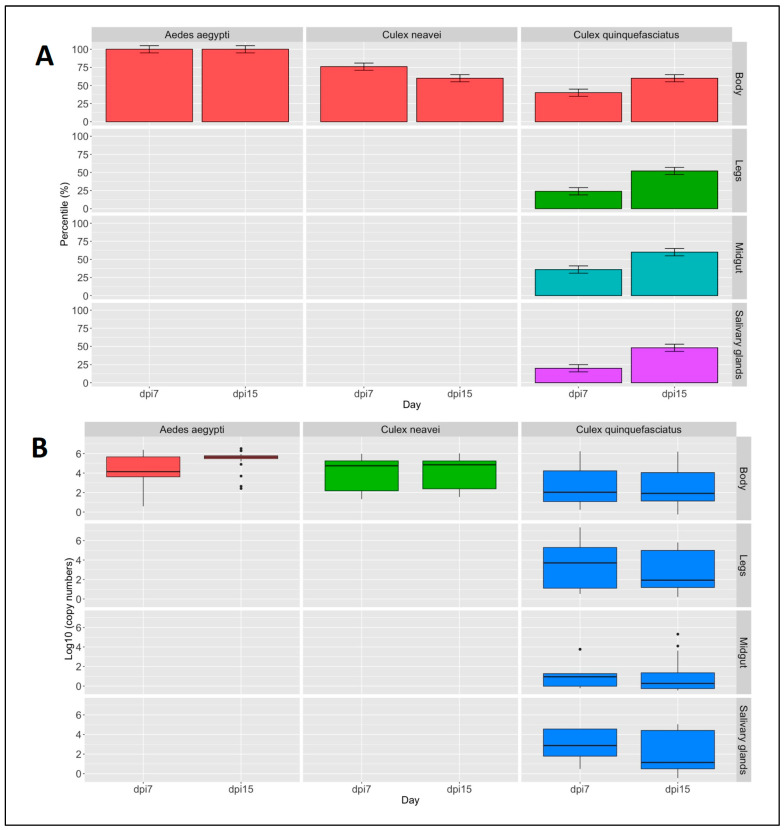
Vector competence of *Aedes aegypti*, *Culex neavei*, and *Culex quinquefasciatus* populations orally exposed to BBKV at 7 and 15 days post-infection (dpi). The (**panel A**) shows the percentage of infection in the body (red) and the percentage of dissemination in legs (green), midgut (blue), and salivary glands (purple) for the three mosquito species. The (**panel B**) shows the corresponding copy numbers detected by the newly established assay from the different organs of *Aedes aegypti* (in red), *Culex neavei* (in green), and *Culex quinquefasciatus* (in blue).

**Table 1 viruses-16-01841-t001:** Description of the newly developed primers and probes.

Names	Primers and Probe	Protein	Nucleotide Position ^a^	CG%	Product Size
FP1	5′-AAACGCAGGAAGAACCAACT-3′	NsP3	5354–5373	45	
RP1	3′-ACCGTCGAAAAGTGATCCGA-5′		5460–5441	45	107 bp
P1	3′-6FAM—TCGTCCGCAGAGCTAGTGCTTG—BHQ1-5′		5381–5402	59.09	

FP: forward primer; RP: reverse primer; P: probe; FAM: fluorescein amidite; BHQ1: black hole quencher 1; NsP3: non-structural protein 3. ^a^ Positions according to the reference sequence NC_001547.1.

**Table 2 viruses-16-01841-t002:** Diagnostic specificity of newly developed qRT-PCR test.

ID	Sample Types	Place of Isolation	Date of Collection	Specie	A.N. or Reference	Viruses Identified	BBKV *q*RT-PCR Assay (Ct)
286219	PM	Barkedji	2016	*C. neavei*	CRORA	Barkedji	Negative
286313	PM	Barkedji	2016	*C. poicilipes*	CRORA	Barkedji	Negative
286109	PM	Barkedji	2016	*C. perfuscus*	CRORA	Barkedji	Negative
286116	PM	Barkedji	2016	*C. neavei*	CRORA	Barkedji	Negative
286085	PM	Barkedji	2016	*C. perfuscus*	CRORA	Barkedji	Negative
286066	PM	Barkedji	2016	*C. neavei*	CRORA	Barkedji	Negative
286126	PM	Barkedji	2016	*C. neavei*	CRORA	Barkedji	Negative
286206	PM	Barkedji	2016	*C. neavei*	CRORA	Barkedji	Negative
288198	PM	Barkedji	2016	*C. poicilipes*	CRORA	Barkedji	Negative
286307	PM	Barkedji	2016	*C. neavei*	CRORA	Usutu	Negative
288129	PM	Barkedji	2016	*C. neavei*	CRORA	Usutu	Negative
286125	PM	Barkedji	2016	*C. neavei*	CRORA	Usutu	Negative
S27 AP	Serum	Uganda	1953	*Human*	AF369024.2	Chikungynya	Negative
ArAAMT/7	Serum	Côte d’Ivoire	1973	*A. africanus*	CRORA	YF	Negative
SH328056	Serum	Senegal	2020	*Human*	MZ513007	RVF	Negative
MR766	Serum	Uganda	1947	*R. monkey*	AY632535.2	Zika	Negative
Dengue 2	Serum	New Guinea	1974	*Human*	AF038403.1	Dengue 2	Negative
Dengue 1	Serum	Cincinnati	1944	*Human*	CRORA	Dengue 1	Negative
ArD76986	Serum	Senegal	1990	*C. poicilipes*	KJ131500	West Nile 1	Negative
ARY168	Serum	Senegal	NA	NA	CRORA	Babanki	25.8
ARMG932	Serum	Madagascar	1984	*C. decens*	CRORA	Babanki	37.3

C: Culex; BBKV: Babanki virus; DK: Dakar; A: Aedes; PM: mosquito pools; AN: accession number; PCR: polymerase chain reaction; AP: African prototype; CRORA: WHO Collaborating Center for Arboviruses and Haemorrhagic Fever Viruses; NA: not available; Ct: threshold cycle.

**Table 3 viruses-16-01841-t003:** Diagnostic sensitivity of new *q*RT-PCR test for BBKV.

ID	Place of Isolation	Specie	Date of Collection	In Vitro Isolation+ IFA	Conventional Pan-Alphavirus RT-PCR	BBKV *q*RT-PCR (Ct)
PM 288115	Barkedji	*C. poicilipes*	2016	SINV	Positive	39.68
PM 288118	Barkedji	*C. neavei*	2016	SINV	Positive	35.9
PM 288121	Barkedji	*C. neavei*	2016	SINV	Positive	36.78
PM 286319	Barkedji	*C. neavei*	2016	SINV	Positive	37.65
PM 286273	Barkedji	*C. ethiopicus*	2016	SINV	Positive	37.35
PM 288064	Barkedji	*C. poicilipes*	2016	Negative	Positive	24.07
PM 286344	Barkedji	*C. poicilipes*	2016	Negative	Positive	39.29
PM 352693	Kedougou	*A. funestus*	2020	Negative	Negative	38.57
PM 352668	Kedougou	*A. argenteopunctatus*	2020	Negative	Negative	38.2
PM 352455	Kedougou	*A. aegypti*	2020	Negative	Negative	36.92
PM 352456	Kedougou	*A. aegypti*	2020	Negative	Negative	35.48
PM 322196	Kedougou	*A. cumminsii*	2020	Negative	Negative	36.9
PM 322195	Kedougou	*A. luteocephalus*	2020	Negative	Negative	36.6
PM 352598	Kedougou	*A. aegypti*	2020	Negative	Negative	37.34
PM 322194	Kedougou	*A. luteocephalus*	2020	Negative	Negative	Negative
PM 286270	Barkedji	*C. poicilipes*	2016	Negative	Negative	Negative
PM 286220	Barkedji	*C. neavei*	2016	Negative	Negative	Negative
PM 352681	Kedougou	*A. furcifer*	2020	Negative	Negative	Negative

C: *Culex*; BBKV: Babanki virus; A: *Aedes*; PM: mosquito pools; PCR: polymerase chain reaction; ID: identification number, Ct: threshold cycle.

## Data Availability

All data are present in this manuscript.

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
