# Peer review of "Establishment of a New Real-Time Molecular Assay for the Detection of Babanki Virus in Africa"

_viruses, 2024, doi:10.3390/v16121841_

Round 1

Reviewer 1 Report

Comments and Suggestions for Authors

Thanks for the opportunity to review this original article.

The authors do an extensive investigation and development of Qrt-PCR system to detect SINV and BBKV with the accuracy of 100%. This is more especially important in the field of medicine and infectious diseases. The control of infectious diseases can only be fought by developing accurate and fast detecting methods. The text is easy to follow and flows with the idea. The graphs, images and tables provided are clear and sited within the text and references are relevance to the work. However, there few issues to be tackled to improve the readability of the manuscript. The authors developed test system is 100% but they did not state any limitation for this test. Below are specific comments.

“Line BBKV is 98% similar” the similarity is based on what parameters?.

Are there current studies on the authors investigated virus. The authors should consider updating the reference list. It contains old references.

How conservative is the genome part used to design primers. These primers could be prone to changes due to mutational in the genome.

What is the fatality rate of BBKV. What biosafety level of the laboratory was used to carry out the experiment? This important information should be discussed in the main text.

Line 191 “to calculated” typo it should be calculate.

In the methods it will improve reading if the author can add a section on statistics and discuss all the tools they used and methods to analyse their data.

The authors should check the consistence of BBKV throughout the manuscript. Sometimes they write babanki virus sometimes they write BBKV for example line 206

Lines 219 to220 are not clear.

Tables: the CT values of higher than 35 are quiet doubtable. This could mean the cross reactivity with primers. What were the CT values for the negative and how did the authors validate the values above 35 to be positive. Some CT values in the Tables are below 35 and the authors write negative for the particular virus they tested and some values are bellow or equal to 35 the authors write positive. The authors should recheck these values and compare them to negative samples.

The authors should check statistics for their experiments. They have to be consistence with what they did. Some parts of the text they mention that data was tested in triplicate and in some text they mention that data was tested in duplicate. If they used both then they should mention and validate that use of duplicate and triplicate.

Author Response

Response to Editor and Reviewer comments: â€¯

The revised manuscript 'Establishment of a new real-time molecular assay for the detection of Babanki virus in Africa' has addressed the majority of the reviewers comments. The following questions remain on this version of the manuscript:

Reviewer #1: The authors do an extensive investigation and development of qRT-PCR system to detect SINV and BBKV with the accuracy of 100%. This is more especially important in the field of medicine and infectious diseases. The control of infectious diseases can only be fought by developing accurate and fast detecting methods. The text is easy to follow and flows with the idea. The graphs, images and tables provided are clear and sited within the text and references are relevance to the work. However, there few issues to be tackled to improve the readability of the manuscript. The authors developed test system is 100% but they did not state any limitation for this test.

  1. “Line BBKV is 98% similar” the similarity is based on what parameters?

Response: We thank the reviewer for this remark. More details have been added to the revised version of the manuscript

  1. Are there current studies on the authors investigated virus. The authors should consider updating the reference list. It contains old references.

Response: We thank the reviewer for this suggestion. However, there are no ongoing studies on the targeted virus. All specific references have been cited in this study. Unfortunately, only the older references provide more specific information on the Babanki virus.

  1. How conservative is the genome part used to design primers. These primers could be prone to changes due to mutational in the genome. â€¯

Response: We thank the reviewer for this comment. More details have been added to the revised version of the manuscript on choice of the targeted region.

  1. 4. What is the fatality rate of BBKV. What biosafety level of the laboratory was used to carry out the experiment? This important information should be discussed in the main text.

Response: Babanki virus is a Biosafety level 2 pathogen as Sindbis virus and details have been added in the revised version of the manuscript.

  1. Line 191 “to calculated” typo it should be calculate.

Response: This was a typo and has been corrected in the revised version of the manuscript.

  1. In the methods it will improve reading if the author can add a section on statistics and discuss all the tools they used and methods to analyse their data.

Response: a section on Statistical and data analysis has been added on the revised manuscript

  1. The authors should check the consistence of BBKV throughout the manuscript. Sometimes they write babanki virus sometimes they write BBKV for example line 206

Response: This has been corrected in the revised version of the manuscript.

  1. Lines 219 to220 are not clear

Response: This section was edited on the revised manuscript

  1. Tables: the CT values of higher than 35 are quiet doubtable. This could mean the cross reactivity with primers. What were the CT values for the negative and how did the authors validate the values above 35 to be positive? Some CT values in the Tables are below 35 and the authors write negative for the particular virus they tested and some values are bellow or equal to 35 the authors write positive. The authors should recheck these values and compare them to negative samples.

Response:  Based on the probit analysis, the assay cut-off for the Ct value is 39.8 (Figure 1B). more details have been added to the head of the Table 3 for better understanding. The column with positive/negative show data from a conventional pan-alphavirus RT-PCR used as reference.

  1. The authors should check statistics for their experiments. They have to be consistence with what they did. Some parts of the text they mention that data was tested in triplicate and in some text, they mention that data was tested in duplicate. If they used both then they should mention and validate that use of duplicate and triplicate.

Response: The specificity analysis has been done by testing in duplicate while the sensitivity test was performed in triplicate for all the dilutions. More details have been added to the revised manuscript.

Reviewer 2 Report

Comments and Suggestions for Authors

Nowadays, the alphvirus detection is almost made with serological test. The aim of the autors is to develop a system of detection by real time PCR of of a subtype of Sindbis virus called Babanki virus (BBKV). Contrary to Sindbis, this virus is reported in northern Europe and Africa.

The system was design after BBKV alignement on NSP3 gene

To test sensitivity and specifity, authors have used in vitro RNA related viruses in Africa area whereas to test sensitivity they selected positive and negative sindbis and babanki virus in serological test from mosquito’ samples. The newly real time PCR system presents 100% specificity.

L25 in the abstract, I don't understand why sensitivity is assessed in comparison with Sindbis. From what I understand, when BBKV exits SINV should not be detected. So it's a question of specificity, not sensitivity.

L169 could the authors correct th qRT-PCR by RT-qPCR due to the quantification during PCR and not RT.

L272 I don't understand what the title of the article has to do with paragraph 3.5: vector competency of mosquitoes for babanki virus. This paragraph confuses the discussion. Shouldn't it be an integral part of an article?

In general, would it be possible to clarify whether this system only detects BBKV or whether it also detects SINV. What is the purpose of this test? I understand that sometimes it detects both (l325) and sometimes it only detects BBKV (l207).

I understand that BBKV is a sub-type of SINV, but please explain why such a system was developed. Please try to be clearer about the use of SINV or BBKV

Author Response

Response to Editor and Reviewer comments: â€¯

Reviewer #2: Nowadays, the alphavirus detection is almost made with serological test. The aim of the authors is to develop a system of detection by real time PCR of a subtype of Sindbis virus called Babanki virus (BBKV). Contrary to Sindbis, this virus is reported in northern Europe and Africa.

The system was design after BBKV alignment on NSP3 gene

To test sensitivity and specificity, authors have used in vitro RNA related viruses in Africa area whereas to test sensitivity they selected positive and negative sindbis and babanki virus in serological test from mosquito’ samples. The newly real time PCR system presents 100% specificity.

  1. L25 in the abstract, I don't understand why sensitivity is assessed in comparison with Sindbis. From what I understand, when BBKV exits SINV should not be detected. So it's a question of specificity, not sensitivity.

Response: We thank the reviewer for this comment. The sentence has been corrected in the revised version of the manuscript.

  1. L169 could the authors correct the qRT-PCR by RT-qPCR due to the quantification during PCR and not RT.

Response: This has been corrected in the revised version of the manuscript.

  1. L272 I don't understand what the title of the article has to do with paragraph 3.5: vector competency of mosquitoes for babanki virus. This paragraph confuses the discussion. Shouldn't it be an integral part of an article?

Response: The vector competency has been used to assess the sensitivity of the assay for detection of Babanki virus in various organs of mosquitoes and its usefulness for experimental studies. More details have added to the revised version of the manuscript.

  1. In general, would it be possible to clarify whether this system only detects BBKV or whether it also detects SINV. What is the purpose of this test? I understand that sometimes it detects both (l325) and sometimes it only detects BBKV (l207).

I understand that BBKV is a sub-type of SINV, but please explain why such a system was developed. Please try to be clearer about the use of SINV or BBKV

Response: The aim of this assay is the specific detection of Babanki virus. Edits have been made on the revised version of the manuscript. More details have been added in the methods and discussion sections for better understanding.